# Analysis of Polymer-Ceramic Composites Performance on Electrical and Mechanical Properties through Finite Element and Empirical Models

**DOI:** 10.3390/ma17153837

**Published:** 2024-08-02

**Authors:** Kiran Keshyagol, Shivashankarayya Hiremath, Vishwanatha H. M., P. Krishnananda Rao, Pavan Hiremath, Nithesh Naik

**Affiliations:** 1Department of Mechatronics, Manipal Institute of Technology, Manipal Academy of Higher Education, Manipal 576104, India; kiran2.mitmpl2022@learner.manipal.edu (K.K.); ss.hiremath@manipal.edu (S.H.); 2Survivability Signal Intelligence Research Center, Hanyang University, Seongdong-gu, Seoul 04763, Republic of Korea; 3Department of Mechanical and Industrial Engineering, Manipal Institute of Technology, Manipal Academy of Higher Education, Manipal 576104, India; vishwanatha.hm@manipal.edu (V.H.M.); pk.rao@manipal.edu (P.K.R.); nithesh.naik@manipal.edu (N.N.)

**Keywords:** permittivity, empirical models, capacitive model, polymer composites, FE analysis

## Abstract

Polymer and ceramic-based composites offer a unique blend of desirable traits for improving dielectric permittivity. This study employs an empirical approach to estimate the dielectric permittivity of composite materials and uses a finite element model to understand the effects of permittivity and filler concentration on mechanical and electrical properties. The empirical model combines the Maxwell-Wagner-Sillars (MWS) and Bruggeman models to estimate the effective permittivity using Barium Titanate (BT) and Calcium Copper Titanate Oxide (CCTO) as ceramic fillers dispersed in a Polydimethylsiloxane (PDMS) polymer matrix. Results indicate that the permittivity of the composite improves with increased filler content, with CCTO/PDMS emerging as the superior combination for capacitive applications. Capacitance and energy storage in the CCTO/PDMS composite material reached 900 nF and 450 nJ, respectively, with increased filler content. Additionally, increased pressure on the capacitive model with varied filler content showed promising effects on mechanical properties. The interaction between BT filler and the polymer matrix significantly altered the electrical properties of the model, primarily depending on the composite’s permittivity. This study provides comprehensive insights into the effects of varied filler concentrations on estimating mechanical and electrical properties, aiding in the development of real-world pressure-based capacitive models.

## 1. Introduction

The design of polymer matrix composites with tailored dielectric properties is crucial for various technological applications. A critical aspect of this endeavor is the ability to predict the effective permittivity of the composite material. Researchers have explored various empirical and theoretical models to achieve this goal. For composites with randomly distributed fillers, the Bruggeman model offers a versatile approach, as demonstrated in studies on Polyvinylidene fluoride (PVDF)-BT [1]. However, limitations arise at higher filler concentrations, prompting the use of alternative models like Maxwell-Garnett, suitable for high Multiwalled Carbon Nano Tubes (MWCNT) content in epoxy resins [2]. The trend in research extends beyond just permittivity, several studies emphasize the importance of considering filler-matrix interactions and interfacial effects. The Kerner model accounts for such interactions in polyimide-graphene oxide composites [3], while the Lichtenecker model with interfacial correction incorporates interfacial polarization in PDMS-CCTO systems [4]. Beyond classical mixing theories, researchers are exploring more sophisticated techniques. The Effective Medium Theory (EMT) captures the composite microstructure’s behavior in BT-Ni composites. The Similar study conducted is, the Finite Element Method (FEM) simulates electric field distribution for complex geometries; Multiscale modeling approaches, combining microscopic and macroscopic techniques offer a promising avenue as demonstrated for Polyaniline (PANI)-clay nanocomposites; Artificial neural networks (ANNs) have emerged as powerful tools for material design; Sigmoidal fitting function for predicting the dielectric constant of BT composite thick films with varying BT content [5]; Empirical mixture laws for predicting the dielectric permittivity of polymer-ceramic composites for energy storage applications [6]; Computer-Aided Design (CAD) using real material morphology for micro granular sample generation to predict effective dielectric properties [7]; Ji et al. [8] employed ANNs to predict the dielectric constant of various polymer-filler combinations for electrical energy storage applications.

Classical models like Maxwell-Garnett (MG), Bruggeman, and Lichtenecker offer advantages depending on the filler concentration. Maxwell-Garnett is suitable for low filler concentrations, as applied to PVDF-PZT or BT composites [9,10,11]. Lichtenecker is suitable for high filler concentrations but ignores particle interactions [12], and Lichtenecker-Rother Model for predicting complex permittivity of polymer composites considering filler cluster aggregation [13]. FEM has been used to analyze dielectric constant, breakdown strength, and microstructure effects in PDMS-BT composites [14]. Additionally, FEM combined Monte-Carlo methods have been used to compare effective permittivity and tangent loss with analytical models for FEM combined with Monte-Carlo methods has been used to compare effective permittivity and tangent loss with analytical models for polymer composites with a carbon-based system [15].

This study aims to develop a modeling approach for polymer-ceramic composites that considers the combined effect of dielectric permittivity and interfacial polarization using the Bruggeman model and Maxwell-Wagner-Sillars (MWS) interfacial polarization, respectively. PDMS will be used as the base material, with BT and CCTO ceramics as fillers. The modeled composite material has been analyzed for its performance in a capacitive pressure sensor application, focusing on capacitance, energy storage capacity, and displacement of the diaphragm of sensor. Furthermore, the interaction of filler particles depending on their distance from each other has been studied by developing a 2D model of the composite material. Using this 2D model, the influence of the distance between particles on electrical distribution and polarization throughout the material has been investigated. This research has the potential to predict the interplay between dielectric permittivity and polarization effects.

## 2. Materials and Methods

The FEM analysis of dielectric composites using empirical models gives a more comprehensive analysis of material performance. In this study polymer-ceramic composite material is modeled using well-established empirical models. For effective dielectric permittivity, various empirical models for the composite material have been analyzed. Using the finite element method, a 3D model of a capacitive pressure sensor (CPS) is modeled. Empirical model of effective dielectric material assigned to the dielectric material property. Uniform pressure is applied on the diaphragm of CPS and analyzed CPS performance for its capacitance, and energy storage capacity of the newly modeled material, also using the 2D model, the interaction of filler material is studied for various electrical parameters.

### 2.1. Empirical Models for Binary Composites

Effective medium theory (EMT) is a method used to estimate the effective permittivity of composite materials. EMT provides a way to approximate the macroscopic properties of the composite based on the properties and proportions of the individual components. The most used models and approaches within EMT are, The Maxwell Garnett model is used for composites where spherical inclusions are embedded in a host material [16], and the Bruggeman model treats the composite as a mixture where both constituents are considered symmetrically [17]. It is particularly useful when there is no clear distinction between the host and the inclusions [18], and the Looyenga model is another approach that can be used when the composite is a random mixture of two phases [19].
(1)$$\varepsilon_{c}=V_{m}\varepsilon_{m}+V_{f}\varepsilon_{f}\;\;\;\;\;\;\;\;\;\;\;(\mathrm{Parallel mixing rule})$$
(2)$${\left(\varepsilon_{c}\right)}^{-1}=V_{m}{\left(\varepsilon_{m}\right)}^{-1}+V_{f}{\left(\varepsilon_{f}\right)}^{-1}\;\;\;\;\;\;\;\;\;\;\;(\mathrm{Series mixing rule})$$
(3)$$\log\left(\varepsilon_{c}\right)=V_{m}log(\varepsilon_{m})+V_{f}log(\varepsilon_{f})\;\;\;\;\;\;\;\;\;\;\;(\mathrm{Logarithmic mixing rule})$$
(4)$$\varepsilon_{c}=\varepsilon_{m}+3V_{f}\left(\varepsilon_{f}-\varepsilon_{m}\right)/\left(\varepsilon_{m}+2\varepsilon_{f}\right)\;\;\;\;\;\;\;\;\;\;\;(\text{Maxwell-Garnett})$$
(5)$$V_{f}\frac{\varepsilon_{f}-\varepsilon_{c}}{\varepsilon_{f}+2\varepsilon_{c}}+V_{m}\frac{\varepsilon_{m}-\varepsilon_{c}}{\varepsilon_{m}+2\varepsilon_{c}}=0\;\;\;\;\;\;\;\;\;\;\;(\mathrm{Bruggeman})$$
(6)$$\frac{\varepsilon_{c}-\varepsilon_{m}}{\varepsilon_{c}+2\varepsilon_{m}}=V_{f}\frac{\varepsilon_{f}-\varepsilon_{m}}{\varepsilon_{f}+2\varepsilon_{m}}\;\;\;\;\;\;\;\;\;\;\;(\text{Clausius-Mossotti})$$where ε_c_, ε_f_, and εm are the dielectric constants of a composite, filler, and polymer matrix material, respectively. V_f_ and V_m_ are the volume fractions of filler and matrix materials. Equation (1) represents the parallel mixing rule, which is an upper bound for the effective permittivity. Equation (2) shows the series mixing rule, which is a lower bound for the effective permittivity, meaning the actual value may be slightly higher due to interactions between filler particles. This model is most applicable for layered or well-aligned filler structures, where the electric field varies across the layers. For random or spherical filler geometries, Equation (3) represents the logarithmic mixing rule, and Equation (4) represents the Maxwell-Garnett model for spherical inclusions, which is generally applicable for low filler concentrations, and it assumes minimal interaction between the filler particles, Bruggeman model is given by Equation (5). Equation (6) shows the Clausius-Mossotti model considers the effect of polarizability effect of matrix and filler inclusions for the calculation of effective dielectric of composites [20].

### 2.2. Empirical Model for the Present Study

There exists a large literature on the empirical models considering few parameters make a less effective predictive model. Developing an empirical model for composite dielectrics by considering permittivity, interfacial polarization, filler size and distribution and elastic modulus makes a more effective predicting model.

#### Effective Dielectric Constant ($\varepsilon_{c})$

The Bruggeman model is a well-established theoretical framework used to estimate the effective dielectric constant of a composite material composed of two distinct phases. This model is particularly useful for estimating the properties of heterogeneous materials where the phases are interspersed in a random manner. The model assumes that each component’s contribution to the overall properties of the composite can be treated symmetrically. The assumptions made for this model are as follows. The two phases are distributed randomly within the composite, the composite material is isotropic, meaning its properties are the same in all directions, the inclusions are assumed to be small enough that their interactions can be neglected, and Both phases contribute symmetrically to the effective properties. The effective dielectric constant of the composite can be modeled using the Bruggeman asymmetric model for a two-phase system is given by Equation (7).
(7)$$\left(\frac{\varepsilon_{f}-\varepsilon_{c}}{\varepsilon_{f}+2\varepsilon_{c}}\right)V_{f}+\left(\frac{\varepsilon_{m}-\varepsilon_{c}}{\varepsilon_{m}+2\varepsilon_{c}}\right)\left(1-V_{f}\right)=0$$

Solving for $\varepsilon_{c}$
(8)$$\varepsilon_{c}=\left(\frac{V_{f}\left(\varepsilon_{f}+2\varepsilon_{m}\right)+\left(1-V_{f}\right)\left(\varepsilon_{m}+2\varepsilon_{f}\right)}{V_{f}\left(3\varepsilon_{m}+\varepsilon_{f}\right)+\left(1-V_{f}\right)\left(3\varepsilon_{f}+\varepsilon_{m}\right)}\right)\varepsilon_{m}\varepsilon_{f}$$

### 2.3. Interfacial Polarization

Interfacial polarization is a crucial phenomenon that significantly impacts the dielectric properties of BT@PDMS/CCTO@PDMS composites. It arises due to the accumulation of charges at the interface between the high permittivity BT/CCTO filler particles and the low permittivity PDMS matrix. This charge accumulation occurs because of the mismatch in permittivity and the inability of mobile charges to completely follow the rapidly alternating electric field at high frequencies [21]. The effective empirical model considering Bruggeman and MWS model is defined as Equation (9).
(9)$$\varepsilon_{\mathrm{ffective}}=\varepsilon_{c}+\frac{KV_{f}\left(1-V_{f}\right){\left(\varepsilon_{f}-\varepsilon_{m}\right)}^{2}}{{\left(V_{f}\varepsilon_{m}+\left(1-V_{f}\right)\varepsilon_{f}\right)}^{2}+\omega^{2}\tau^{2}{\left(V_{f}\varepsilon_{m}+\left(1-V_{f}\right)\varepsilon_{f}\right)}^{2}}$$
where, $\varepsilon_{c}$ is the dielectric constant obtained from the Bruggeman model, which balances the contributions from the filler and the matrix. K is the shape factor depending on filler geometry, ω is the angular frequency of the applied electric field, τ is the relaxation time. The MWS effect introduces a polarization term that depends on the geometry of the filler, volume fraction of the filler, and the difference in dielectric constants between the filler and the matrix.

### 2.4. Modeling a Capacitive Pressure Sensor

To evaluate the electrical and mechanical performance of the proposed composite materials (BT@PDMS and CCTO@PDMS), the CPS model was utilized. COMSOL Multiphysics software (6.0) was employed to model a cylindrical capacitive pressure sensor, as depicted in Figure 1a. The sensor has a radius of 20 mm and consists of three layers. The top and bottom layers serve as electrodes for the electrical supply, while the middle layer is comprised of the dielectric composite material (BT@PDMS or CCTO@PDMS). The thickness of the dielectric layer was set to 5 mm. Initial conditions of zero charge on the sensor and zero initial pressure on the diaphragm were assumed. A potential difference of 1 volt was applied across the electrodes, and a uniform pressure was applied on the sensor’s diaphragm. Material properties relevant to the simulation are provided in Table 1. Mesh optimization was conducted by testing different mesh shapes and element sizes. The triangular mesh elements were identified as the optimal choice (as shown in Figure 1b) [22].

The capacitance of the model can be calculated using Equation (10).
(10)$$C=\varepsilon_{r}\varepsilon_{0}\cdot \frac{A}{d}$$
where C represents the capacitance in Farads (F), ε_r_ represents the relative permittivity of the dielectric composite material (BT@PDMS or CCTO@PDMS), $\varepsilon_{0}$ represents the permittivity of free space, A represents the area of the parallel electrodes in square meters (m^2^), and d represents the thickness of the dielectric layer in meters (m). The total energy stored in the capacitive model is given by Equation (11).
(11)$$W=\frac{1}{2}CV^{2}$$
where, W represents the total electrical energy stored in Joules (J), V represents the potential difference (voltage) applied across the electrodes in Volts (V).

### 2.5. Modeling of Composite Material

The interaction of inclusion particles is crucial for analyzing electrical parameters such as electric field distribution and polarization effects. A 2D model is created using a multiphysics FEM tool. It consists of a PDMS matrix material with BT inclusion particles uniformly dispersed within it. In this case, perfectly spherical shape inclusions are considered. To study the electrical parameters, a potential of 1 Volt is applied across the material, as shown in Figure 2a. Zero initial voltage and charge on the material are assumed for this study. A free triangular mesh was applied to the model, perfectly converging at all the edges, as depicted in Figure 2b. Inclusion particles are dispersed in the matrix such that they are equidistant from each other.

For the particle interaction analysis, a square slab of size 10 × 10 µm with 0.42 µm diameter circular particles is created. In this 2D model, the distance between the centers of two particles, defined as “a”, is varied. The distances analyzed are 7.5, 3.5, and 0.5 times the diameter of the particle, as well as the scenario where the particles are just touching each other. The effect of these varying distances on the composite material has been studied.

## 3. Results and Discussion

The empirical and finite element modeling approaches have been adopted to analyze the electrical and mechanical performance of composite materials. In the empirical method, the particle size effect was neglected to estimate the composite’s permittivity. Figure 3a,b illustrate the empirical results of the composite’s permittivity with varied concentrations of BT and CCTO in the PDMS material. The proposed empirical model was compared with the series and parallel empirical models to determine the boundaries of the composite’s permittivity. It was observed that the proposed model is within the boundary of the conventional models, and the filler concentration increases with an increase in the permittivity of the BT/PDMS and CCTO/PDMS composite materials. Additionally, CCTO/PDMS has a higher effective permittivity than BT/PDMS because CCTO has a higher permittivity constant.

The electrical performance of the composite material is evaluated based on the permittivity obtained from the proposed empirical model. The capacitance of the composite was estimated using the finite element modeling approach and was observed to increase linearly with an increase in filler concentration, as shown in Figure 4a. The CCTO composite exhibited higher capacitance than the BT composite, with a maximum of 854.29 nF and 533.85 nF at a 40% volume fraction of CCTO and BT in PDMS, respectively. Similarly, the electric energy stored with varied volume percentages of BT and CCTO fillers is shown in Figure 4b. The CCTO/PDMS composite can store more energy than the BT/PDMS material, which corresponds to the higher capacitance of the CCTO composite. There was twice as much energy storage in the CCTO composite compared to the BT composite in the capacitive model. Therefore, the CCTO composite is a more promising material for energy storage applications.

The mechanical performance of the capacitive model under varied pressure was evaluated for the CCTO and BT composites. Absolute strain energy was developed in the capacitive model upon applying different pressure loads with varied filler concentrations, as shown in Figure 5a,b. The strain energy, resulting from load pressure on the composite-based capacitive model, leads to elastic energy. As the pressure increases, the strain developed in the capacitive model also increases, and this is also influenced by the concentration of filler content in the composite material. the BT variation exhibited a higher strain rate compared to the CCTO material, likely because of the modulus of elasticity of the CCTO material.

Further, stress developed in the capacitive model under varied pressure and filler content of the CCTO and BT composites is shown in Figure 6a,b. The uniaxial pressure in the capacitive model leads to slight variations in stress with different volume fractions. There was a sudden rise in stress at 10%, possibly because the model considered it a complete solution in the concentrated area of the BT material. A similar effect was observed in the CCTO material at 40%. Additionally, the stress developed in the BT variation is greater compared to the CCTO variation in the captive model, likely because the BT/PDMS composite material properties are less stiff than those of the PDMS/CCTO composite material.

Continuing with that, displacement is a crucial parameter in the capacitive model, mainly dependent on the model’s geometry and the material’s permittivity. The displacement with varied volume percentages of the filler under different pressures for BT and CCTO concentrations is shown in Figure 7a and Figure 7b, respectively. In the case of BT variation, there is an initial rise in displacement which then flattens with more BT concentration. This is because the 10% increase in stress leads to greater displacement, depending on how the model optimally resolves the solution. Similarly, for CCTO variation, an improvement in displacement is noticed from 30% filler concentration onwards. Thus, the initial concentration has less influence on the model’s displacement. Comparatively, CCTO filler content shows more displacement than the BT filler content, indicating that CCTO/PDMS composite material properties are a more influential factor in the capacitive model.

Further studies were conducted on the interaction between filler and matrix materials and its effect on electrical parameters. A finite element analysis of a 2D composite model was performed to estimate the potential distribution, electric charge, and polarization by varying the filler ratio within the matrix material. The BT filler material, with a particle size of 0.42 µm and spherical shape, was considered. The filler concentration in the matrix materials was increased based on the particle size radius. The gap between the particles in the matrix was set at 7.5 times, 3.5 times, 0.5 times, and touching each other’s radius. The voltage distribution across the composite structure was observed. It was noted that voltage disperses more easily through the composite with a large gap between the filler particles compared to a densely packed filler composite. Additionally, as filler concentration increases, the voltage distribution across the composite slows down. Thick particles result in more controlled voltage distribution within the composite, as shown in Figure 8a–d.

Furthermore, the polarization effect between the particles in the matrix material is illustrated in Figure 9a–d. It was observed that due to the applied voltage, all particles in the matrix become polarized regardless of their concentration level. The polarization effect is more pronounced at lower filler content than at higher filler content. Consequently, the polarization disperses near the filler particles, increasing the attraction force and reducing the dipole movement between the particles. Therefore, modeling the interaction between the matrix and filler helps to understand the dipole movement and polarization effect in the composite material. This estimation is crucial for predicting the electric effect between dissimilar materials before conducting experimental studies.

Additionally, the electric field distribution between the particles in the matrix with varied filler content levels is shown in Figure 10a–d. The electric charge distribution is more apparent at lower filler content than at higher filler levels. As the concentration level increases, so does the electric field. The charge dipole movement indicates how the filler in the matrix material holds the charge and attempts to distribute it to adjacent filler material. Notably, as the distance between filler particles decreases to the point where they touch, the electrified field doubles compared to the composite with particles spaced 7.5 times apart. Closely packed particles create a proper network between the filler, leading to an increase in the electric field, as demonstrated in the 2D model of the filler and matrix interaction examined.

In the end, electric displacement aids the movement of charge between the filler and matrix, as shown in Figure 11a–d. Charge displacement is minimal when particles are spaced apart. To enhance the displacement effect between the matrix and filler, closely packed filler particles influence charge movement more effectively. Thus, a higher filler content forms chain-like structures, where one filler pulls another, leading to electric field displacement in the composite material. Additionally, electric displacement is closely related to the applied electric field and the material’s permittivity. Therefore, materials with higher permittivity exhibit greater electric displacement. Improvement in electric displacement results in higher degrees of polarization and dipole movement. This modeling approach has elucidated the electric phenomena of filler and matrix interaction, depending on the material’s permittivity. Therefore, studying the interfacial effects between filler and matrix provides a more comprehensive understanding of the composite’s electrical behavior.

Continuing with the present study, recent work on modeling dielectric materials using different composite materials needs to be explored. Various fillers, matrices, and empirical approaches have been examined in this context. Table 2 provides information on modeling the dielectric properties of polymer composites. The literature summarized in the table highlights the importance of different empirical approaches for understanding composite behavior. By utilizing the diverse modeling toolbox offered by established techniques and advanced tools like Artificial Neural Networks, researchers can select the most suitable model for a specific scenario. Factors such as filler concentration and particle interaction become key considerations when choosing between models like Maxwell-Garnett (ideal for low concentrations) or Bruggeman/Lichtenecker (better for moderate/high concentrations). This study takes a unique approach by combining existing permittivity and electric modulus models, potentially leading to a more comprehensive understanding. This combined approach, along with the exploration of various modeling techniques, underscores the importance of selecting the appropriate tool for estimating the composite material behavior.

## 4. Conclusions

The present study highlights different empirical approaches to estimating the dielectric permittivity of polymer-based composites, specifically PDMS/BT and PDMS/CCTO composites. Based on the dielectric permittivity of the polymer composite, a capacitive model was implemented to understand the effect of filler concentration on the capacitance and energy storage in the model. Additionally, a 2D model was formulated to analyze how varying filler concentration levels in the matrix material influence electric properties. The study found that the proposed empirical model can estimate the permittivity of the composite within acceptable limits for both BT and CCTO filler content. The CCTO filler composite exhibited higher permittivity than the BT composite. Furthermore, capacitance and energy storage in the model increased with higher filler concentrations, indicating that the CCTO/PDMS composite material is particularly suitable for capacitance applications. Similarly, the mechanical features showed consistent effects in both composites under varying pressure, with stress, strain, and displacement being key influencing factors in the capacitive model. Moreover, the interaction of filler variation in the polymer matrix demonstrated significant impacts on voltage distribution, polarization, electric field, and electric displacement. Increased filler concentration improved the permittivity of the composite, which in turn enhanced the electric and mechanical properties of the model. Thus, this study provides a comprehensive understanding of filler and matrix interactions and their application in capacitive models, offering valuable insights for future practical case studies on pressure-based capacitive models.

## Figures and Tables

**Figure 1 materials-17-03837-f001:**
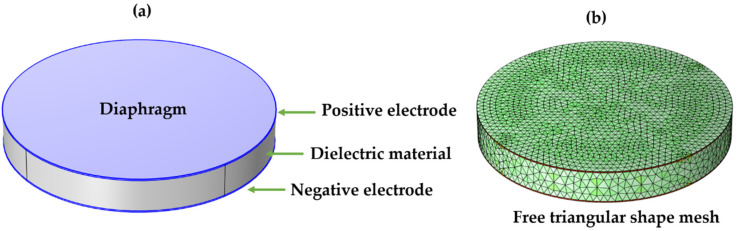
Capacitive pressure sensor mode. (**a**) 3D model of sensor, (**b**) Free triangular mesh applied to model.

**Figure 2 materials-17-03837-f002:**
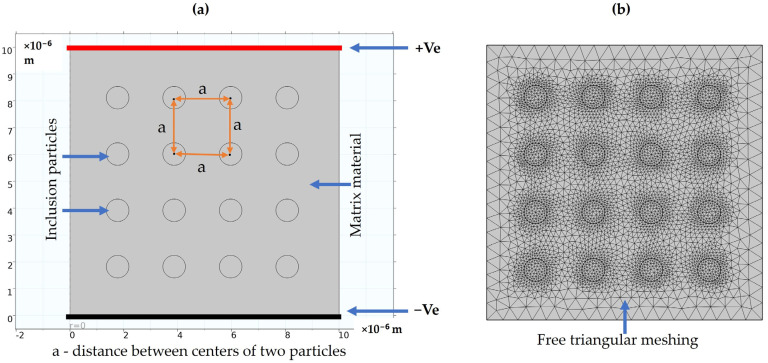
2D model of composite material. (**a**) Particle and matrix interfacial model, (**b**) Free triangular mesh application to the model.

**Figure 3 materials-17-03837-f003:**
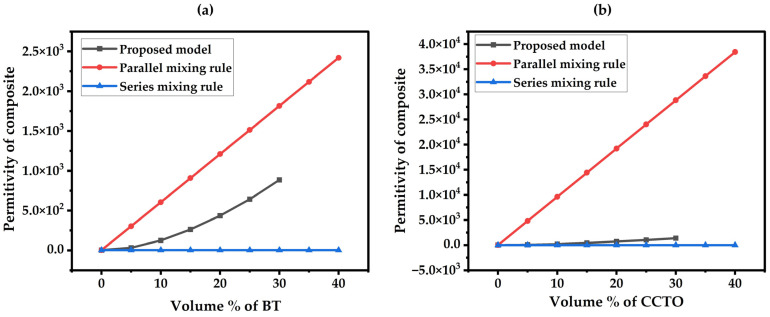
Permittivity of empirical models. (**a**) For BT@PDMS Composites, (**b**) For CCTO@PDMS Composites.

**Figure 4 materials-17-03837-f004:**
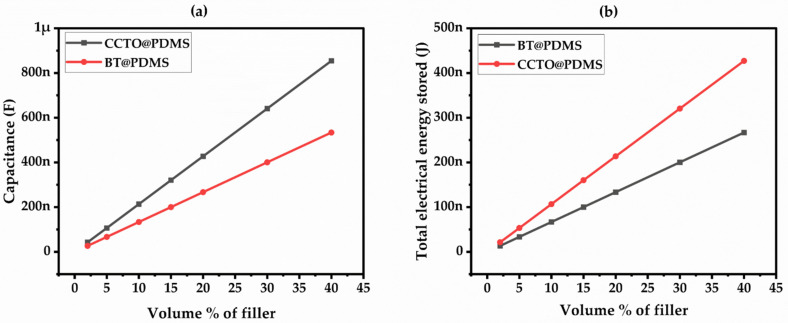
(**a**) Capacitance with a varied volume percentage of BT and CCTO, (**b**) Total Electrical energy stored with a varied volume percentage of BT and CCTO.

**Figure 5 materials-17-03837-f005:**
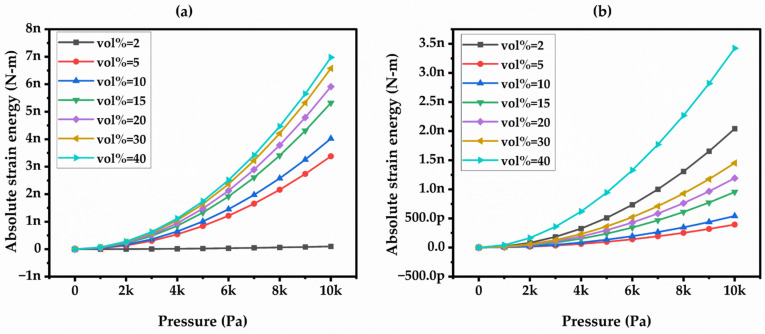
Absolute strain energy vs. pressure in (**a**) BT Filler and (**b**) CCTO Filler Composites.

**Figure 6 materials-17-03837-f006:**
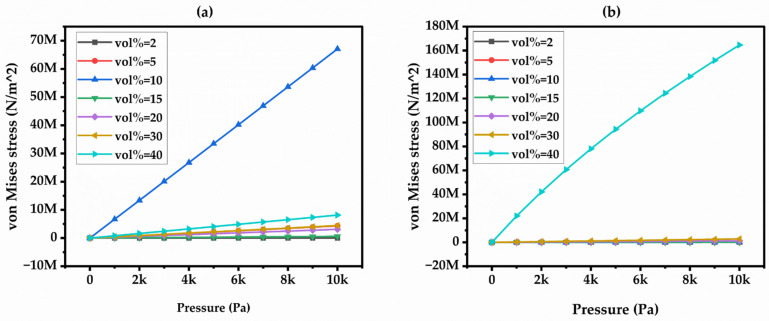
von Mises Stress vs. Pressure in (**a**) BT Filler and (**b**) CCTO Filler Composites.

**Figure 7 materials-17-03837-f007:**
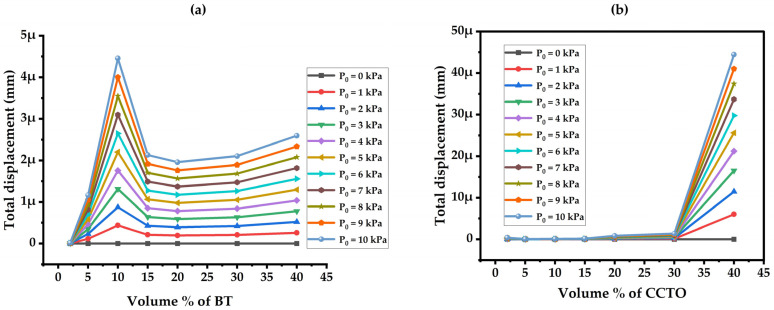
Total Displacement vs. Volume % Filler. (**a**) BT Filler and (**b**) CCTO Filler Composites.

**Figure 8 materials-17-03837-f008:**
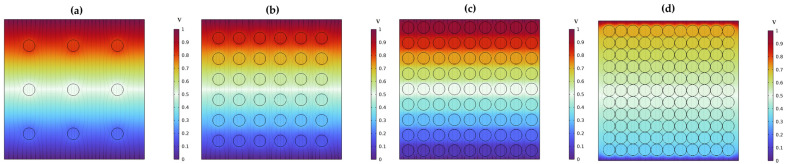
The voltage distribution, when BT filler particles are embedded in a base material: (**a**) spacing between the particles is 7.5 times the radius, (**b**) 3.5 times the radius, (**c**) 0.5 times the radius, and (**d**) particles are just touching each other.

**Figure 9 materials-17-03837-f009:**
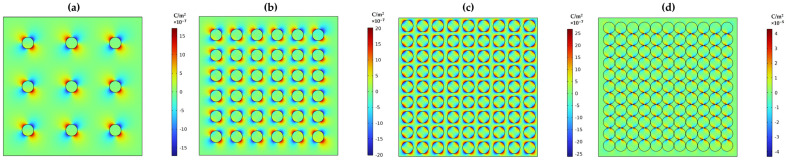
Polarization response to particle spacing in a base material (**a**) spacing between the particles is 7.5 times the radius, (**b**) 3.5 times the radius, (**c**) 0.5 times the radius, and (**d**) particles are just touching each other.

**Figure 10 materials-17-03837-f010:**
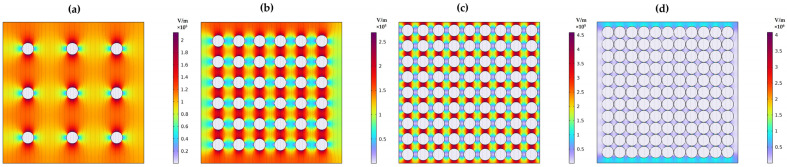
Electric field distribution response of particle spacing in a base material (**a**) spacing between the particles is 7.5 times the radius, (**b**) 3.5 times the radius, (**c**) 0.5 times the radius, and (**d**) Particles are just touching each other.

**Figure 11 materials-17-03837-f011:**
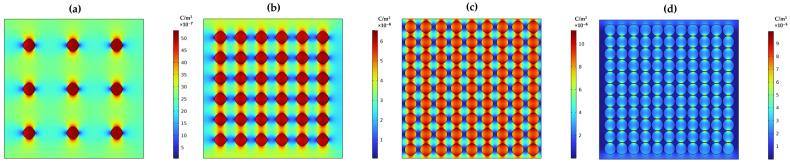
Electric displacement field response to particle spacing (BT fillers) a base material: (**a**) spacing between the particles is 7.5 times the radius, (**b**) 3.5 times the radius, (**c**) 0.5 times the radius, and (**d**) particles are just touching each other.

**Table 1 materials-17-03837-t001:** Materials and materials properties.

Materials	Permittivity (ε)	Young’s Modulus (E) [GPa]	Poisson’s Ratio (ν)	Density [kg/m^3^]	Particle Size [µm]	Ref.
PDMS	2.4	0.00075	0.45	970	-	[23]
BT	6000	60	0.25	6000	0.42	[24,25]
CCTO	9600	256	0.25	4700	0.72	[25,26]
Silver	1.971	78	0.37	10.49	-	[27]

**Table 2 materials-17-03837-t002:** Comparative study on preexisting empirical models for dielectric composites.

Reference	Matrix	Filler	Empirical Model	Remarks
[1]	PVDF	BT	Bruggeman	Predicts permittivity for random BT distribution
[2]	Epoxy Resin	MWCNTs	Maxwell-Garnett	Applicable for high MWCNT concentrations
[3]	Polyimide	Graphene Oxide	Kerner	Accounts for filler-matrix interaction
[4]	PDMS	Calcium Copper Titanate oxide (CCTO)	Lichtenecker with Interfacial Correction	Considers interfacial polarization effects
[28]	BT	Nickel (Ni)	Effective Medium Theory (EMT)	Captures behavior of composite microstructure
[29]	Poly(methyl methacrylate) (PMMA)	Silver Nanowires	FEM	Simulates electric field distribution for complex geometries
[30]	Polyaniline (PANI)	Clay Nanoparticles	Multiscale Modeling (Combining Models)	Combines microscopic and macroscopic approaches
[31]	High-Density Polyethylene (HDPE)	BT Microspheres	Logarithmic Mixing Rule with Percolation Threshold	Simple model considering filler connectivity
[32]	Poly(lactic acid) (PLA)	Cellulose Nanofibers	Mori-Tanaka Model	Considers the effect of fiber orientation
[33]	Polycarbonate (PC)	Exfoliated Graphite Flakes	Dielectric Mixture Theory with Maxwell-Wagner-Sillars (MWS)	Captures interfacial effects at high frequencies
[34]	Benzocyclobutene (BCB)	Silicon Dioxide (SiO_2_) Nanoparticles	Self-Consistent Mean Field (SCMF) Theory	Powerful method for complex composites
[8]	Various Polymers	Various (potentially with shells)	Artificial Neural Network (ANN)	Predicts dielectric constant for material design in electrical energy storage
[35]	Bisphenol A diglycidylether Epoxy	Hollow Ceramic Spheres	Effective Medium Theory (Clausius-Mossoti relation)	Bisphenol A diglycidylether Epoxy
[5]	X7R BT Particles	BT Gel	Sigmoidal Fitting Function (based on Boltzmann equation)	Predicts dielectric constant of BT composite thick films with varying BT gel content.
[36]	Epoxy Resin (RE)	BT	Empirical Mixture Laws (unspecified)	Predicts dielectric permittivity of polymer-ceramic composites for energy storage applications
[7]	Barium Strontium Titanate (BST)	Magnesium Borate	CAD Approach.	Predicts effective dielectric properties of binary ceramic composites with high dielectric contrast.
[37]	Poly(vinylidene fluoride) (PVDF)	Lead Zirconium Titanate or BT	Maxwell-Garnett Ignores particle interactions.	Suitable for low filler concentrations.
			Bruggeman Accounts for particle interactions to some extent.	Suitable for moderate filler concentrations.
			Lichtenecker Ignores particle interactions	Suitable for high filler concentrations.
[38]	Polymer (ε′ = 2.73, ε″ = 0.2)	Aluminum powder	Nielsen’s equation	Predicts dielectric properties of metal-insulator composites for metal-insulators.
[6]	PDMS	CB	Various Models	Predicts dielectric permittivity
[39]	PDMS	CB	Wiener Model fits best	Predicts dielectric permittivity of PDMS/CB nanocomposites.
[40]	Various Polymer	Ceramic, Magnetic, CB	Modified Lichtenecker-Rother Model Accounts for aggregation of filler particles	Predicts complex permittivity of polymer composites considering filler cluster aggregation.
[41]	PDMS	BT	FEM	Analyzes dielectric constant, breakdown strength, and microstructure effects.
[42]	Polymer with permittivity (ε) = 3.02	BT	FEM and Monte-Carlo methods	Compares effective permittivity and tangent loss with analytical models.
[43]	Lead Zirconate Titanate based composites with various microstructures		Combining Landau-Devonshire theory and FEM	proposes a valuable modeling tool for designing materials with tailored permittivity.
[44]	Epoxy Resin	BT Particles	FEM with quasi-static approximation	Analyzes dielectric constant using periodic FCC structures with novel packing protocols.

## Data Availability

Data is contained within the article.

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
