# Peer review of "Analysis of Polymer-Ceramic Composites Performance on Electrical and Mechanical Properties through Finite Element and Empirical Models"

_materials, 2024, doi:10.3390/ma17153837_

Round 1

Reviewer 1 Report

Comments and Suggestions for Authors

Review of the manuscript entitled ‘Innovative Modeling of Polymer-Ceramic Composites for Enhancing Electrical and Mechanical Properties through Finite Element and Empirical Analysis

Authors present a modeling approach for polymer-ceramic composites that considers the combined effect of both dielectric permittivity and elastic modulus. PDMS is used as the base material, with BT and CCTO ceramics as fillers. The modeled composite material is then analyzed for its performance in a capacitive pressure sensor, focusing on displacement, capacitance, and energy storage capacity.

The study is interesting and overall this is a thoroughly investigated work that is likely to appeal to the readership of the journal ‘materials’. However, I have some remarks to improve the quality of the manuscript and its understanding.

- p 3 l 116: authors don’t explain the goal of Effective Medium Theory (EMT).

- p4 l 135 Vfi is not given.

- p7 l 254: ‘he near-linear increase in capacitance…’

- p7, l 259: The interfacial effects between the filler and the PDMS matrix might also play a role in influencing the overall permittivity and capacitance. How do you consider this interfacial effect? How do you modelized it? Is it a local permittivity?

- p9 l 312: multiphysics softewre

- How random distribution of filler nanoparticles influence the results?

- p11 l 377: Permittivity influences how a material stores electrical energy in an electric field. The sentence is repeated.

- p 12 l 428: By incorporating these interfacial effects, the analysis would provide a more comprehensive understanding of the composite's electrical behavior. How do you plan to do that?

- p14 l 459 : the sentence ‘exploring multiscale modeling for a deeper understanding of structure-property relationships in these composites’ is not clear. What do you mean?

Author Response

Response to Reviewer 1

Dear Sir/Madam,

Thank you for giving us the opportunity to submit a revised draft of our manuscript titled ‘Innovative Modeling of Polymer-Ceramic Composites for Enhancing Electrical and Mechanical Properties through Finite Element and Empirical Analysis’. We appreciate the time and effort that you have dedicated for providing your valuable feedback on the manuscript. We are grateful to you for your insightful comments on the paper. We have been able to incorporate changes to reflect most of the suggestions provided by you.

Comments and Suggestions for Authors: Authors present a modeling approach for polymer-ceramic composites that considers the combined effect of both dielectric permittivity and elastic modulus. PDMS is used as the base material, with BT and CCTO ceramics as fillers. The modeled composite material is then analyzed for its performance in a capacitive pressure sensor, focusing on displacement, capacitance, and energy storage capacity.

The study is interesting and overall this is a thoroughly investigated work that is likely to appeal to the readership of the journal ‘materials. However, I have some remarks to improve the quality of the manuscript and its understanding.

  1. p 3 l 116: authors don’t explain the goal of Effective Medium Theory (EMT).

Response: Thank you for the valid suggestion. To improve readability the paragraph has been modified without changing its original meaning. In the updated version of the manuscript your suggestion is implemented from line no. 92 to 113.

  1. - p4 l 135 Vfi is not given.

Response: Thank you for your observation. For the simple explanation we have removed that equation because next equation is defining the same parameters (Eq 8).

  1. - p7 l 254: ‘he near-linear increase in capacitance…’

Response: Thank you for finding typo error. The typo error has been corrected (We have re-written to increase readability)

  1. - p7, l 259: The interfacial effects between the filler and the PDMS matrix might also play a role in influencing the overall permittivity and capacitance. How do you consider this interfacial effect? How do you modelized it? Is it a local permittivity?

Response: The interfacial effects between the filler and the PDMS matrix significantly influence the overall permittivity and capacitance of the composite. The properties of the filler and polymer composite material were estimated using an empirical model. Subsequently, the capacitive model was analyzed with the effective parameters derived from this empirical model. The interface effects were considered as a collective model with independent variations in filler content to analyze permittivity and capacitance.

  1. - p9 l 312: multiphysics softewre

Response: Thank you for finding spelling mistake. Mistake has been rectified

  1. - How random distribution of filler nanoparticles influence the results?

Response: The random distribution of filler nanoparticles significantly influences the electrical and mechanical properties in the finite element (FE) model. Random distribution can lead to variations in local density, creating regions with differing electrical fields, polarization, and charge distribution as well as mechanical strain, stress, and displacement. This heterogeneity can affect the overall dielectric permittivity and capacitance, as well as the stress-strain behavior of the composite material. By simulating different random distributions, we can better understand how filler dispersion impacts the performance and reliability of the composite, providing valuable insights for optimizing material design and application.

  1. - p11 l 377: Permittivity influences how a material stores electrical energy in an electric field. The sentence is repeated.

Response: Thank you for the detailed study and observation. The repeated sentence is removed in the updated manuscript by retaining same meaning.

  1. - p 12 l 428: By incorporating these interfacial effects, the analysis would provide a more comprehensive understanding of the composite's electrical behavior. How do you plan to do that?

Response: To incorporate interfacial effects and provide a more comprehensive understanding of the composite's electrical behavior. Further, it can be plane that, Utilize finite element modeling techniques that can capture the detailed interactions at the interface between the filler nanoparticles and the PDMS matrix. This includes considering the influence of particle size, shape, and distribution on the electric field distribution and polarization within the composite. Then, apply specific boundary conditions at the interface regions to accurately simulate the charge accumulation and potential barrier effects. This will involve defining appropriate electrical and mechanical boundary conditions that reflect the real interfacial interactions. Finally, Conduct experiments to measure the electrical properties of composites with known filler distributions. Compare these experimental results with the FE model predictions to validate and refine the model, ensuring it accurately reflects the real interfacial effects. Therefore, by integrating these strategies, we aim to enhance the fidelity of our FE models in representing the interfacial effects, leading to a deeper and more precise understanding of the composite's electrical behavior.

- p14 l 459 : the sentence ‘exploring multiscale modeling for a deeper understanding of structure-property relationships in these composites’ is not clear. What do you mean?

Response: Thank you for pointing that out. We agree that the sentence was unclear in this context. As such, it has been removed from the revised manuscript.

Reviewer 2 Report

Comments and Suggestions for Authors

This work developed a model for the combined effect of permeability and elastic modulus on the properties of polymer-ceramic composites, exemplified on PDMS with barium titanate and calcium copper titanate as fillers. The model has the advantage that it can be applied at high ceramic filler content, prone for use in high capacitance and energy storage applications. It is an entirely theoretical approach, based on simulations.

The Abstract can stand alone for understanding the achievements of the work. The Introduction is focused on the theories and models employed to explore/ predict the materials behaviour.

In Materials and Methods section, the authors explain clearly how the composite materials are modelled. The Results and Discussion section includes the discussion of different empirical models for permittivity and the reason for why specific model is selected for further simulations. Also, a comparison was performed for capacitance, total electrical energy stored, Von Mises stress, total displacement versus filer %vol, in case of two composites: PDMS-barium titanate and PDMS-calcium copper titanate. The results show the superiority of PDMS- calcium copper titanate composite material. The interaction of filler particles between them and between particles and matrix was explored for the composite PDMS-BT. A mini review of empirical models for dielectric composites is offered in Table 2. The table highlights criteria for selecting the most suitable model in different situations.

In the Conclusions section, the authors stress on their contribution to the design of novel composite dielectric materials and highlight the importance of considering the interplay permittivity-elasticity in pressure sensors design. They intend to validate experimentally their model in the future.

Minor corrections in the text are needed:

-          PDMS polymer- polydimethylsiloxane. Write the complete name before its abbreviation at first appearance in the text (in the Abstract).

-          The same for the other abbreviations: PVDF, MWCNT, PMMA (in Introduction section).

-          Lines 58-63: the senteces lack predicates. I suggest putting them in a phrase, separated by semicolons. E.g.” Other theories employed to explore were:..;...;...”

-          Equation 2: vm and vf must be written in capital letters unless they represent others than Vm and Vf.

-          Define φ in Eq.9.

Author Response

Response to Reviewer 2

Dear Sir/Madam,

Thank you for giving us the opportunity to submit a revised draft of our manuscript titled ‘Innovative Modeling of Polymer-Ceramic Composites for Enhancing Electrical and Mechanical Properties through Finite Element and Empirical Analysis’. We appreciate the time and effort that you have dedicated for providing your valuable feedback on the manuscript. We are grateful to you for your insightful comments on the paper. We have been able to incorporate changes to reflect most of the suggestions provided by you.

Comments and Suggestions for Authors:

This work developed a model for the combined effect of permeability and elastic modulus on the properties of polymer-ceramic composites, exemplified on PDMS with barium titanate and calcium copper titanate as fillers. The model has the advantage that it can be applied at high ceramic filler content, prone for use in high capacitance and energy storage applications. It is an entirely theoretical approach, based on simulations.

The Abstract can stand alone for understanding the achievements of the work. The Introduction is focused on the theories and models employed to explore/ predict the materials behaviour.

In Materials and Methods section, the authors explain clearly how the composite materials are modelled. The Results and Discussion section includes the discussion of different empirical models for permittivity and the reason for why specific model is selected for further simulations. Also, a comparison was performed for capacitance, total electrical energy stored, Von Mises stress, total displacement versus filer %vol, in case of two composites: PDMS-barium titanate and PDMS-calcium copper titanate. The results show the superiority of PDMS- calcium copper titanate composite material. The interaction of filler particles between them and between particles and matrix was explored for the composite PDMS-BT. A mini review of empirical models for dielectric composites is offered in Table 2. The table highlights criteria for selecting the most suitable model in different situations.

In the Conclusions section, the authors stress on their contribution to the design of novel composite dielectric materials and highlight the importance of considering the interplay permittivity-elasticity in pressure sensors design. They intend to validate experimentally their model in the future.

Minor corrections in the text are needed:

  1. PDMS polymer- polydimethylsiloxane. Write the complete name before its abbreviation at first appearance in the text (in the Abstract).

Response: Thank you for the valid suggestion. The care has been taken in the updated manuscript to implement your comment.

  1. The same for the other abbreviations: PVDF, MWCNT, PMMA (in Introduction section)

Response: Thank you for the valid suggestion. The care has been taken in the updated manuscript to implement your comment.

  1. Lines 58-63: the senteces lack predicates. I suggest putting them in a phrase, separated by semicolons. E.g.” Other theories employed to explore were:..;...;...”

Response: Thank you for the valid suggestion. Your suggestion is incorporated in the updated manuscript. (Line no. 48-58)

  1. Equation 2: vm and vf must be written in capital letters unless they represent others than Vm and Vf.

Response: Thank you for the valid suggestion. The care has been taken in the updated manuscript to implement your comment.

  1. Define φ in Eq.9.

Response: Thank you for the valid suggestion. φ is defined in terms of Vf and defined in the updated version of manuscript.

Reviewer 3 Report

Comments and Suggestions for Authors

The manuscript deals with an FEM modelling of the changing strain energy and dielectric permittivity in polydimethylsiloxane-ceramic composites using the example of the two fillers BT and CCTO. A few comments:

1. The authors mention that the chosen filler concentration range of 5-40% is below the percolation threshold and in particular does not allow to reach the plasticity region or to observe the non-linear region of strain energy change. Why did the authors not choose higher filler concentrations?

2. How was the pressure applied? Was it uniaxial and omnidirectional pressure?

3. It is necessary to add information on how the strain energy and the rate of strain energy were determined.

4. The reasoning in lines 274-275, 283-288 that “...higher stiffness slower increase in strain energy..”, “.. higher stress under pressure due to its reduced deformation...”, “.. This limited deformation concentrates the stress within the material, leading to a higher overall stress state”, “..more even stress distribution and potentially higher overall stress...” are not so unambiguous and need additional justification.

5. What is considered a total displacement? What is meant by a higher or lower total displacement? Does the polymer matrix become more brittle or is this the result of an increase in its stiffness? The role of stiffness and plasticity in changes in strain energy needs to be clarified. The possible reasons given for the different mechanical response of composites are too general and are not limited to those listed. Which ones are decisive?

6. Fig. 6-9: Is it possible to predict which filler concentration corresponds to the stated distances between the particles?

Comments on the Quality of English Language

I noticed a few typos

Author Response

Response to Reviewer 3

Dear Sir/Madam,

Thank you for allowing us to submit a revised draft of our manuscript titled ‘Innovative Modeling of Polymer-Ceramic Composites for Enhancing Electrical and Mechanical Properties through Finite Element and Empirical Analysis’. We appreciate the time and effort that you have dedicated to providing your valuable feedback on the manuscript. We are grateful to you for your insightful comments on the paper. We have been able to incorporate changes to reflect most of the suggestions provided by you.

Comments and Suggestions for Authors: The manuscript deals with an FEM modelling of the changing strain energy and dielectric permittivity in polydimethylsiloxane-ceramic composites using the example of the two fillers BT and CCTO.

A few comments

  1. Comments: The authors mention that the chosen filler concentration range of 5-40% is below the percolation threshold and in particular does not allow to reach the plasticity region or to observe the non-linear region of strain energy change. Why did the authors not choose higher filler concentrations?

Response: Response: Thank you for pointing out this statement. In the revised manuscript, the sentence has been removed and appropriate inferences have been mentioned. Regarding the filler concentration, while higher filler concentrations can be used during modeling, practical cases often deviate from expected results due to issues such as agglomeration and insolubility. Thus, based on the available literature, we restricted the filler content to 40% in the modeling case.

  1. Comments: How was the pressure applied? Was it uniaxial and omnidirectional pressure?

Response: Response: Thank you for the important point. The capacitive model was created using Multiphysics software, and the respective properties were assigned to analyze its pressure effect on capacitance and energy storage. The pressure was applied under boundary load conditions with uniaxial pressure.

  1. Comments: It is necessary to add information on how the strain energy and the rate of strain energy were determined.

Response: Thank you for your comment. In the revised manuscript, we have included detailed information on how the strain energy and the rate of strain energy were determined.

(During the application of pressure load on the circular capacitive model, it undergoes deformation from its initial position. The strain rate and strain energy of the model were derived from the structural parameters of the finite element software used in the analysis. Specifically, varying filler concentrations and pressure loads allowed us to record distinct strain rates and strain energies, providing a comprehensive understanding of their effects on the capacitive model. This approach helps in better understanding the strain effects within the capacitive model under different experimental conditions.)

  1. Comments: The reasoning in lines 274-275, 283-288 that “...higher stiffness slower increase in strain energy..”, “.. higher stress under pressure due to its reduced deformation...”, “.. This limited deformation concentrates the stress within the material, leading to a higher overall stress state”, “..more even stress distribution and potentially higher overall stress...” are not so unambiguous and need additional justification.

Response: Thank you for your feedback. The sentences in question have been removed from the manuscript. In the revised version, we have provided clearer justifications and explanations to better support our findings

  1. Comments: What is considered a total displacement? What is meant by a higher or lower total displacement? Does the polymer matrix become more brittle or is this the result of an increase in its stiffness? The role of stiffness and plasticity in changes in strain energy needs to be clarified. The possible reasons given for the different mechanical response of composites are too general and are not limited to those listed. Which ones are decisive?

Response: Thank you for bringing up these points. In the capacitive model, total displacement refers to the overall movement observed under varying pressure conditions and different filler concentrations. The displacement is higher towards the center of the model and lower towards its boundaries. It's important to clarify that displacement in this context is influenced by the composite material's characteristics rather than its stiffness or plasticity. Therefore, in the revised manuscript, we have provided clearer explanations to better elucidate the displacement behavior observed in the capacitive model.

  1. Comments: 6-9: Is it possible to predict which filler concentration corresponds to the stated distances between the particles?

Response: Thank you for your question. In the present study, we used the gap between particles to represent the concentration level of filler content in the polymer material. It is indeed possible to predict the filler concentration based on features such as the radius of the particles and the overall area of the composite. In future studies, we plan to investigate how varying filler ratios in the polymer material can improve dielectric values. This will help us further understand the relationship between filler concentration and composite properties.
